# Recruiting and Engaging Women of Reproductive Age with Obesity: Insights from A Mixed-Methods Study within A Trial

**DOI:** 10.3390/ijerph192113832

**Published:** 2022-10-24

**Authors:** Sarah Louise Killeen, David F. Byrne, Aisling A. Geraghty, Cara A. Yelverton, Douwe van Sinderen, Paul D. Cotter, Eileen F. Murphy, Sharleen L. O’Reilly, Fionnuala M. McAuliffe

**Affiliations:** 1UCD Perinatal Research Centre, School of Medicine, University College Dublin, National Maternity Hospital, D02 YH21 Dublin, Ireland; 2UCD Institute of Food and Health, School of Agriculture and Food Science, University College Dublin, D04 V1W8 Dublin, Ireland; 3APC Microbiome Ireland, Biosciences Research Institute, National University of Ireland, T12 K8AF Cork, Ireland; 4School of Microbiology, National University of Ireland, T12 K8AF Cork, Ireland; 5Teagasc Food Research Centre, Moorepark, Fermoy, P61 C996 Cork, Ireland; 6Precision Biotics Ltd. (Novozymes Cork), Cork Airport Business Park, Kinsale Road, T12 D292 Cork, Ireland

**Keywords:** mixed-methods, SWAT, qualitative, recruitment, obesity, women’s health

## Abstract

Engaging women with obesity in health-related studies during preconception is challenging. Limited data exists relating to their participation. The aim of this study is to explore the experiences and opinions of women participating in a weight-related, preconception trial. This is an explanatory sequential (quan-QUAL) mixed-methods Study Within A Trial, embedded in the GetGutsy randomized controlled trial (ISRCTN11295995). Screened participants completed an online survey of eight questions (single or multiple choice and Likert scale) on recruitment, motivations and opinions on study activities. Participants with abdominal obesity (waist circumference ≥ 80 cm) were invited to a subsequent semi-structured, online focus group (n = 2, 9 participants) that was transcribed and analyzed using inductive thematic analysis, with a pragmatic epistemological approach. The survey (n = 102) showed the main research participation motivations were supporting health research (n = 38, 37.3%) and wanting health screening (n = 30, 29.4%). Most participants were recruited via email (n = 35, 34.7%) or social media (n = 15, 14.7%). In the FGs, participants valued flexibility, convenience and. research methods that aligned with their lifestyles. Participants had an expanded view of health that considered emotional well-being and balance alongside more traditional medical assessments. Clinical trialists should consider well-being, addressing the interconnectedness of health and incorporate a variety of research activities to engage women of reproductive age with obesity.

## 1. Introduction

Obesity is the most common health condition in women of reproductive age [1]. Young adults are at risk of weight gain for many reasons including the impact of managing multiple priorities and major life transitions during this phase of their life-course [2]. Globally, body mass index (BMI)-classified obesity prevalence is rising and is estimated to be greater than 20% in all women by 2025 [3,4]. Despite this, women aged 18–35 years are largely underrepresented in obesity research [5,6]. Reaching the right population group is essential for the efficacy of health interventions and insights from these populations helps inform appropriate intervention delivery [1,2]. Systematic reviews of trials in young adults with overweight or obesity show that often only small sample sizes are achieved, and this limits statistical power [7,8]. The problem of low recruitment is further exacerbated by low retention rates among women of reproductive age [9].

Much of the research to date on the motivations, barriers and enablers, and overall preferences of women with overweight or obesity on study conduct has focused on weight loss or maternal health [10,11,12]. This is an issue as weight loss or pregnancy may not be a priority for many young women with obesity, affecting their engagement with these trials [13,14,15,16]. It is acknowledged that focusing on improving the health of all women of reproductive age could help to avoid these issues, yet there is a lack of data on the preferences of women in this group in relation to study participation [14]. This limits our ability to design successful health intervention studies with sufficient quality and representative samples [7,8,9]. Up to 50% of pregnancies are unplanned, so many young women may not consider their reproductive potential in the context of their health [10,16,17]. It is therefore difficult to engage women in the preconception period in health-related programmes [18].

A Study Within A Trial (SWAT) is a self-contained research study embedded in a host trial that can be used to explore factors relating to study design [19]. The GetGutsy study was a randomised controlled trial of a probiotic intervention in women with abdominal obesity who were not pregnant, lactating or planning a pregnancy in the next 3 months. The primary outcome was reduced inflammation without weight loss, diet or lifestyle changes [20]. We designed a SWAT, embedded in GetGutsy to assess the success of the recruitment strategies employed and attitudes towards factors relating to study processes. The aim was to fill the identified research gap on the most effective aspects of study design that will encourage women of reproductive age, particularly women with obesity, to take part in general health-related research.

## 2. Materials and Methods

### 2.1. Study Design

This was a mixed-methods explanatory sequential (quan-QUAL) SWAT using an electronic survey and focus groups (FGs) (Figure 1). Integration of the methods occurred during data collection and interpretation [21]. This was a single-center study carried out from November 2019 to May 2020 at the University College Dublin (UCD) Perinatal Research Centre and affiliated with the National Maternity Hospital, a tertiary University Hospital for maternity services in Dublin, Ireland. The quantitative survey was circulated electronically in November-December 2019, with a reminder sent out in March 2020 [19]. FG participants were recruited via email or telephone. The FGs took place online via Zoom in April 2020 and May 2020. The study was conducted according to the guidelines of the Declaration of Helsinki and approved by the Institutional Review Board of University College Dublin and the National Maternity Hospital in 2017 (EC 28.2017) and updated in 2019 (EC 28.2017). 

### 2.2. Host Trial

The GetGutsy study (ISRCTN11295995) was a randomised controlled trial of a probiotic intervention versus placebo in women with overweight or obesity and evidence of metabolic dysregulation (HDL cholesterol < 1.29 mmol/L and/or triglyceride concentration ≥ 1.7 mmol/L) [20]. Women were eligible for inclusion in the trial if they were aged 18–65 years, English speaking, not pregnant, lactating or planning a pregnancy in the next three months, had a Body Mass Index (BMI) ≥ 28 kg/m^2,^ a waist circumference ≥ 80 cm, and were not planning health behaviour changes aimed at weight loss in the next 3 months. Although limited in the context of defining obesity, BMI was used as a practical way to identify potential participants who had excessive adipose tissue that affected health in relation to lipid profile. The BMI cut off ≥ 28 kg/m^2^ for initial population-wide recruitment was chosen instead of BMI 30 kg/m^2^ due to research suggesting that most women with overweight or obesity may underestimate their BMI, and the increased risk associated with higher waist circumference [22,23]. General eligibility (including age and BMI) was assessed through self-reported data received by phone or email. Where appropriate, women were invited to attend in-person screening to confirm eligibility through demographics, anthropometry, and a blood test. At the first study visit, these measures were repeated and additional data on diet and lifestyle was collected data before randomisation. The trial primary outcome was to determine the impact of a *Bifidobacterium* probiotic on high-sensitivity C-Reactive Protein. Traditional recruitment strategies (posters, flyers, and recruitment drives, a radio appearance) were employed in University College Dublin, several local hospitals in Dublin, local businesses, and corporate wellness events. These were supplemented by digital recruitment strategies including direct email through employers or professional organization mailing lists, social media posts, and blog posts. Women were also encouraged to share the study information with their peers. Interested women self-identified to the research team. 

### 2.3. Study Population

Women were eligible to take part in the SWAT survey if they underwent initial basic screening for the GetGutsy trial (height, weight, and age) and consented to future contact (Figure 1). As the aim of the survey was to collect broad insights on aspects of conduct for a study designed for young women with obesity, all women who expressed interest in the study were invited, even if they did not meet the criteria for study participation. As the aim of the focus groups was to gain rich data on the specific aspects of the study as they related to the target population, only women who were potentially eligible to original GetGutsy trial (met all criteria except an atherogenic lipid profile) were invited to take in the SWAT FG. We did not limit the sample size for this study to gain the maximum size possible within the available participant pool from the original GetGutsy trial. 

### 2.4. Online Survey

A cross-sectional, online survey was developed and piloted prior to use [21]. The survey aimed to elicit the general views of women who registered interest in taking part in the trial. The survey comprised of eight questions, covering internal and external factors relating to women’s experiences in the trial over three main areas, (1) participating, (2) motivations for participation, (3) opinions on future study design (Appendix A) [21]. Questions contained a mix of free text, multiple choice, and Likert scale responses. Demographic data was also collected regarding age, height, weight, employment status, highest educational attainment, and previous research participation. The survey underwent content validation following review by experienced researchers from public health, dietetics, nutrition, obstetrics, and midwifery. The survey was circulated via email and completed using Google forms. Data was entered into an excel file (Microsoft Excel® for Microsoft 365 MSO (Version 2209 Build 16.0.15629.20152, Microsoft, New York, NY, USA). Statistical analysis was performed using IBM SPSS software for Windows, version 26.0 (SPSS Inc, Chicago, IL, USA). Chi-square tests were used to compare categorical variables. Significance was determined at *p* < 0.05. The survey results informed the development of the FG topic guide. 

### 2.5. Focus Groups

The FGs aimed to extend the results of the quantitative survey in a smaller group of women with overweight or obesity and explore their lived experience of taking part in research. The COREQ checklist is reported as Appendix A [24]. The focus groups (n = 2) used a pragmatic epistemological approach [25]. Women were allocated to one of two focus groups based on availability. A semi-structured topic guide was developed by SLK, DB and SOR (Appendix A), which was reviewed and piloted with experienced members of the research team for content validation, clarity and potential biases. The topic guide asked about the experience of taking part in the trial including their motivations for volunteering and their experiences of recruitment, data collection, and communication with the researchers. Specific aspects of trial design were also explored such as using BMI as an inclusion criterion. All women invited were provided with an information sheet prior to participating. The FGs (95 and 120 min) were conducted by SLK (female registered dietitian (BSc) and PhD student) and DB who took field notes (male research assistant with a BSc and MPH). Both researchers were experienced in qualitative research, were involved in running the trial, and had met the participants (n = 9) in person at their screening visit prior to the FGs. The focus groups were digitally recorded via Zoom. 

The digital audio recordings were transcribed verbatim. Each participant was given a participant identification number. Any identifying information was removed prior to analysis. Transcripts were not returned to participants and minor stylistic edits were made to some verbatim quotes to aid in reader comprehension including removal of redundant words and verbal utterances [26]. Field notes were also reviewed as part of the analysis. Transcripts were reviewed for familiarisation and analysed independently by two researchers (SLK and DB) using content and inductive thematic analysis [27,28]. SLK coded manually, while DB used NVivo 12 to facilitate sharing [29]. The research team for analysis included SOR (registered dietitian, academic researcher, and associate professor with extensive experience with qualitative research in women’s health) and AAG (nutritionist and academic researcher with experience in women’s health research and qualitative analysis). Researcher triangulation, peer debriefing and team consensus on themes were key activities during the theme definition and refining stages to minimise bias. 

## 3. Results

### 3.1. Characteristics of the Study Population

Over a hundred (n = 102) of the n = 498 invited women answered all survey questions fully and were included in the analysis (Table 1). Just over a quarter (26.5%, n = 27) of survey respondents were below the age of 35 years and 14 (13.7%) had obesity as defined by BMI ≥ 30 kg/m^2^. Most women (n = 38, 37.3%) reported that they wanted to support health research as their motivation for taking part (Table 1). In terms of the focus groups, out of the 60 invitations sent out to all eligible women via email, 15 women agreed to participate and nine attended. A total of two focus groups were conducted (Table 2). Four out of the nine women in the focus groups had children. All had completed some third level education and were of a white ethnicity. Two women had a BMI between 28–30 kg/m^2^ while the remaining seven had a BMI over 30 kg/m^2^ (Table 2).

### 3.2. Quantitative Results

Results on how women were recruited to the study and their preferred methods of recruitment can be seen in Table 1. More women below 35 years put social media in their top 3 methods of study recruitment, χ^2^ = 4.576 (df (1), n = 102, *p* = 0.043), otherwise no differences were seen. The most valued benefit to taking part in research was the opportunity to support health research, which was selected as one of the top three benefits by 66.7% (n = 68). In relation to time taken to contact the research team after hearing about the study, 79.3% (n = 81) of respondents outlined that they contacted the research team within two days. In terms of potential barriers to participation, only 14.7% (n = 15) of women said they would be less likely to take part if a blood sample was required, 26.5% (n = 27) if there was a 50% chance of getting the intervention, 23.5% (n = 24) if their work colleagues knew they were taking part and 38.3% (n = 39) for needing to provide a stool sample. Collectively 45.1% (n = 46) of women agreed or strongly agreed that a weight criterion as part of the eligibility assessment would make them less likely to share the study with friends and 40.2% agreed or strongly agreed it would affect their sharing of it at work (n = 41). 

### 3.3. Qualitative Results

Three main FG themes were identified (Figure 2). These were: “how it makes me feel”, “how it fits into my world” and “understand the whole of me” with the woman at the centre connecting them. 

#### 3.3.1. Theme 1: How It Makes Me Feel

This theme relates to the impact of research on the emotions of participants and how it resonated with their values. Women who identified with the eligibility criteria and took part in the trial reported feelings of altruism as an intrinsic motivation for volunteering in the trial as illustrated by P8. 

*P8: when I see one that’s…me…I go right let’s..take this on*!

Women or reproductive age valued the novelty of the opportunity to learn more about research to increase general awareness of women’s’ health issues amongst their peers, as illustrated below. 


*P3: I was just interested to see how, how studies are done as well; I had never been given the offer of taking part in any kind of study before…*


*P1: …I think women need talk about a lot more about their health…I’m just so happy to share and help with it*.

Word of mouth was highlighted several times as a valuable method to promote the study and engage women, whilst also advocating for women’s health issues, although how appropriate this is may depend on the study design.

*P6: … ask for that referral, ask people to spread the word*.


*P2: I think maybe it does depend on what the study is, …. you’d be more willing to share something personal with your girlfriends …you don’t necessarily want everybody knowing your business …*


Receiving individualised feedback on their health was another driver of participation as it was seen to gain momentum in behaviour changes. 

*P1: ….sometimes you need a bit of a kickstart to get yourselves back on track*.

*P8: …maybe this is something I can do to improve my health even if I can’t take part anymore*.

As highlighted by P2, feedback on outcomes that have a financial cost for assessment, or outcomes that are more difficult to measure autonomously were the most appreciated part of the trial. Knowledge about current health status was considered valuable to inform behaviour changes 


*P2: anything that you know gives me a free check-up, I was like sign me up where do I go?*


Participants in the FG also reported that the host trial and SWAT were minimally invasive, and this affected their interest in taking part, although they acknowledged that there will be some individuals who dislike blood tests and would therefore not participate. 


*P8: Yeah, I guess in previous studies that I had taken part in, they um, because they needed the blood tests and stuff…a good few people in the office who were like, can’t deal with blood, can’t deal with giving blood tests.*


The importance of study location was identified with participants discussing the impact of attending a maternity hospital for their study visits had on them. Specifically, concern around unsolicited questions around their intentions to conceive or gynaecological health were highlighted. This led to an overall discussion around the sensitivity of women’s health such as hormones or fertility, over other health measures such as those included in the trial, as illustrated below.


*P7: … I was a bit like aw I don’t want people to make assumption.*


*P8: … I think the levels of things that you’re talking about are em… not the type of things that could make things awkward…hormone levels or some other type of a health marker that may make things a little bit more of a difficult conversation*.

When discussing the impact that the weight criterion for eligibility had on their comfort in sharing the trial with their peers, the initial response was positive, and women felt it was important to share this as part of the full suite of eligibility criteria. Further discussion highlighted that the level of comfort they had depended on the person with whom the trial was shared, as shown by P3. 

*P3: There’s probably some people that if it was a study tracking weight then I wouldn’t share it with them because I know maybe they’re sensitive about their weight or if they’ve had a bad relationship with their weight in the past*.

Suggestions such as framing the weight criterion in a sensitive nature and providing information on why the weight criterion is included as part of the eligibility criteria may overcome this barrier. 


*P5: I think the use of the phrase non-judgemental would be really important when weight is being used and… this isn’t necessarily about us looking at you as the subject but looking at lots of different women.*


#### 3.3.2. Theme 2: How It Fits into My World

This theme relates aspects of study design that support participation and limit impact on the current lifestyle of the participants. The subtheme of convenience was evident with women valuing the ease of communicating with the researchers and setting up appointments for data collection at times that suited them. Early morning and weekend appointments were valued as this supported participation around work and family commitments. 


*P5: I was able to make appointments …quite accommodatingly early in the morning because then I was able to go to work straight after…I would definitely consider doing another study.*


Preferences for email or telephone communication varied depending on the personal circumstances of the individuals and their unique personalities. 


*P6: you know I’ve stopped looking at emails at the moment, purely because there’s so many coming in from the school…I just can’t read another email.*


*P8: I actually have like a real aversion to picking up the phone at all so I..am a much more of an email person so even if it is three or four back and forth…it’s easy for me to kind of respond during a meeting*.

The perceived convenience of study location also impacted the extent to which the women promoted the trial among their networks. 


*P8: most of my friends….live on the other side of the city! So…I wouldn’t have advertised it much around…my peer group.*


Aspects of the intervention such as the number of study visits, were important in determining whether women volunteered and remained in the study. Specific concerns around incomplete adherence to the intervention were highlighted as potential barriers to participation. Interventions that limit impact on current lifestyles including work, family and socialising were most valued. 

*P5: It didn’t take too much from me you know that I wasn’t having to come in on a weekly basis or anything like that*.


*P1: you’d always be worried if you’re busy in work that you wouldn’t take it on time!*


#### 3.3.3. Theme 3: The Whole of “Me”

This theme relates to how participants identify with the research and how the study considers their needs, preferences, and individuality. The subtheme of the holistic and interconnected nature of health emerged when discussing health outcomes included in the trial. Women reported a desire to incorporate measures relating to their emotional well-being as this was considered as important, if not more important than physical or functional measures. 


*P9: …I think there’s too much of a kind of divide between our body and our minds and I think you know, a healthy mind will lead to a kind of a healthier body….…it’s kind of like we’ve compartmentalised all of these things … for me, it’s about the holistic sense of us, being…*



*P4: …you sometimes just focus on that BMI number, but it is the more holistic view… like it’s not all bad…*


This was relevant to the inclusion of BMI as an eligibility criterion for the study as the usefulness of this as a marker of health was questioned. Rather than using BMI as a defining characteristic, consideration of a variety of measures of health were favoured. Several women highlighted the aspects of their health that were thriving despite their BMI. 


*P8: … I’ve got my blood tests and everything and lung function, all of that kind stuff is fine but just that BMI figure, is just that little over…and like yourselves, you know, you obviously had that as…an entry point to this study!*


Women discussed the variable nature of responses to interventions, particularly in the context of body composition. A lack of response to standard interventions or an inability to apply general information to their personal circumstances were highlighted as barriers to engagement in health behaviours. 


*P3: there’s just so much information out there…it’s hard to know...what you should be doing….*


The subtheme of personalisation was identified from the data. Interventions that are tailored to suit personal health characteristics, likes and dislikes and previous experiences were discussed (P1). Specifically, interventions including time with a coach or clinician were highlighted, as illustrated by P8. 

*P1: I think everyone is so different, aren’t they? …what would work for me mightn’t work for somebody else*.

*P8: that personalisation kind of almost that you know, having that consultant time where somebody really looks at my life*. 

## 4. Discussion

We found women of reproductive age value research methods that align with their lifestyles and maximise convenience. Measures such as blood tests and stool samples were acceptable to this population as part of a clinical trial. Learning about health and giving back were clear motivations for study participation. In the FGs, women valued the unique opportunity to learn about research and themselves, while advancing women’s health. Participants had an expanded view of health that considered emotional well-being and balance alongside more traditional physical and functional assessments when asked what being healthy meant to them. They also acknowledged the variable nature of lifestyles and of responses to health interventions, valuing a person-centred, flexible approach over traditional, defined interventions. Factors that linked with maternity settings or reproduction were considered more sensitive than weight or lipids, highlighting a need for neutral locations and language use in preconception research. 

Women in our study were recruited by, and preferred, digital methods. This is of benefit to clinical trialists. In pregnancy, evidence suggests digital strategies may be more effective and lower cost compared to traditional means [30]. In women planning a pregnancy, higher responses to digital strategies have also been reported [12]. Our study looked at this phenomenon in women of reproductive age, outside of the context of pregnancy. This is important given engagement with health-related material may differ in women based on pregnancy history or intention to conceive [31,32,33]. By exploring attitudes towards the use of digital media in addition to their efficacy, we address a current gap in the literature [34]. We also provide novel data on the response time of women to recruitment strategies. Most women in our study contacted the research team within two days. This finding can be used by clinical trialists to evaluate the impact of their recruitment activities. After digital, the next most effective recruitment method was word of mouth. The importance of this was identified in the FGs. In the WIC study, Di Noia et al. found benefits to word-of-mouth and suggested that asking mothers to refer other participants to aid study recruitment [35]. Our data suggest a combination of these factors may prove useful in women of reproductive age [36]. 

The original GetGutsy study included a weight criterion to identify participants with potential atherogenic lipid profile. In the recent Canadian Adult Obesity Clinical Practice Guidelines, obesity is defined as the presence of excess adiposity that negatively impacts health [37]. Using this definition, a raised BMI alone is not considered obesity. Applying this definition to metabolic health alone would remove the metabolically healthy obesity phenotype, leaving only “healthy” or “obesity”. Some studies, however, suggest that there is risk of adverse outcomes in obesity based on BMI even without any evidence of metabolic derangement [38,39,40]. Others have suggested that metabolically healthy obesity is a transient state, during which individuals with obesity based on BMI are at increased risk of developing metabolic derangements overtime [41,42,43]. Taken together, this data suggests that regardless of metabolic status, it is important that people with raised BMI have access to treatment should they wish to receive it and clinical trialists should consider this in future study design [44].

We found just over 50% of survey respondents reported wanting a health screen or to be healthier as their main motivation for registering interest in the GetGutsy study. It is long documented in the literature that people living with obesity may have less engagement with health screening for a variety of factors, including weight stigma [45,46,47,48]. While our survey was completed by women regardless of BMI, we also found an interest in health screening in our focus groups, which included only women with overweight or obesity. In a recent study, other factors such as education, age and location of residence were associated with missed health screenings [49]. It is possible that sociodemographic factors explain the favourable view of health screening in our study. Our study was not about weight loss and as such, recruitment content focused on health screening including lipid analysis rather than weight. The qualitative component of this SWAT identified an interest in learning new things about health amongst women with overweight and obesity. Future clinical trials and health services can consider the value of novelty to frame health screenings.

Inclusion of a weight criterion in a study to identify potential participants with obesity, as defined by adiposity-related ill-health, may affect the comfort women have in sharing a study with some of their peers and family members. Our survey found over 40% of women or reproductive age, from a variety of BMI categories, were less likely to share a study with a weight criterion. Inviting women to take part in a trial with a weight-criterion could result in weight-stigma [50]. Weight stigma may reduce the desire to engage with a research study as it would require the participant to identify with an externally stigmatised population group [2,51]. Others have found a reluctance from healthcare professionals in discussing weight if they feared it would be negatively received by patients [52,53]. Despite this, the ACTION-IO study of 1500 people with BMI-classified obesity in the UK found only 16% of their cohort did not feel comfortable discussing weight [54]. In pregnancy, Mills et al. found women with obesity reported a negative experience when their weight was avoided by healthcare professionals in their antenatal care [55]. Further discussion of this topic in our FGs identified that while comfort sharing a study with a weight criterion may vary depending on the relationship or perceived comfort discussing weight or the other party, women with obesity did not see this as a barrier for study recruitment. In addressing weight, FG participants highlighted the need for transparency and sensitivity in communication about weight with research participants. Our results provide further insight and nuance into recent research suggesting that there is a disconnect between the perceived offence with discussions on weight by intervention delivery agents and people living with obesity, including women of reproductive age [53,54]. This suggests that trials and clinicians should consider mentioning addressing but not focusing on weight alone. This will provide people with obesity with the information they need whilst reducing weight stigma.

Measurement of a wider range of health outcomes has been suggested as a method to reduce weight stigma in the clinical setting [56]. We found when discussing health and weight, women with abdominal obesity felt greater importance should be placed on emotional well-being as an influencer of health and behaviours. The importance of a holistic approach in obesity has been highlighted previously, with weight loss referred to as a “mental battle” [57]. The recent qualitative study by Timmermans et al. found that women with obesity before, during, or after pregnancy thought mental well-being should be addressed before lifestyle changes [58]. Even in the context of a weight management intervention, young women with obesity have a preference study names that focused on health, fitness, and well-being, rather than weight loss [59]. The work by Ogden et al. suggests a shift towards non-quantified outcomes in relation to obesity such as improved fitness or a change in relationship with food [60]. The desire for consideration of the interconnectedness of health found in our study suggests the importance of well-being is not limited to weight loss but should be considered in general health-related trials for women of reproductive age with obesity. 

To encourage engagement, clinical trialists working with women of reproductive should consider how the research meets the needs of their population and aligns with their personal interests. During preconception, a lack of knowledge is commonly identified as a barrier to behaviour change [61]. In pregnancy, Murray-Davis et al. found the lack of availability of information was a key issue for participating mothers, especially after pregnancy [62]. Our study builds on this data, highlighting women of reproductive age participating in a health-related trial desire more information to support behaviour change. We also found women preferred personalised and flexible interventions. This in line with previous research [57,58,63]. Hutchesson et al. found young women reported an interest in participating in weight loss programs that are specifically designed for them [64]. The need for flexibility of interventions has been found in several studies including a variety of communication options with intervention deliverers, through in person and online means [59,65]. Our study suggests these intervention characteristics are important across a broad range of studies for women of reproductive age and are not limited to weight loss or maternity care. 

Finally, we found study location is an important factor in study participation [11]. Crino et al. reported similar findings whereby location convenience, defined by considerations such as access to parking and distance from work, was important [59]. In addition to convenience, the characteristics of the study location may be an important consideration. The location of the study, in this case a maternity hospital, was identified as a potential barrier to participation as it may invite unsolicited questioning around intention to conceive or gynaecological health. The mixed-methods SWAT by Kozica et al., found concern around lack of anonymity and self-consciousness as barriers to participation in lifestyle studies [66]. Fear has also been highlighted as a psychosocial factor that could influence study participation [63]. Our study provides novel insight into the impact maternal health-related outcomes may have on these factors. We found hormones and topics surrounding maternal health were considered as more sensitive than weight. The workforce could as a neutral and convenient location for preconception research, that could run in conjunction with employee wellness programs [67,68,69,70]. 

### Strengths and Limitations

Our study provides novel and pragmatic data on aspects of the attitudes and preferences of women of reproductive age in relation to study design with a focus on obesity. Unlike previous research, the focus is general research in women’s health and does not pertain exclusively to weight loss or maternal health trials. The integration of qualitative and quantitative data in this study supports the translation of findings to the target population [71]. The approaches used in this study can be applied to other trials [19]. This study used an explanatory sequential mixed methods approach, which is well suited to answer this specific research question [21]. 

There are also some limitations worth noting. Participants in our study were all interested in taking part in health-related research, so it is not possible to generalise the results to all women. The women in our study were all White, had completed at least some third level education and were most over 35 years. Future trials could consider incorporating of a similar SWAT to explore whether these factors differ in more heterogenous populations, including younger women and incorporating it into their study processes at recruitment to capture factors associated with screening disinterest. Most respondents were in full, or part time employment and the findings may therefore not be generalisable in other circumstances. Aspects of the survey design may have influenced responses including that the survey was circulated online only, and that we provided pre-defined options for most questions. It is possible that women completing the online survey may prefer this method of communication more than those choosing to not engage in the survey, therefore potentially inflating the level of preference. Regardless, the findings towards this preference are in line with the literature [34]. The limitations of susceptibility to observer and interviewer bias for the FG should be noted. We did not include any patient or public involvement in this study, or the original GetGutsy. Future research would benefit from early involvement of patient-advocacy groups and other stakeholders to gain further insight into factors influencing study participation along with survey validation. This would enhance our confidence in the questions being asked and their acceptability and importance to the target audience. Finally, our study did not have an *a priori* sample size calculation as participants were already limited to those who were either interested in, or potentially eligible for, the original GetGutsy trial. Future studies would benefit from incorporating SWAT protocols to provide more insight into what works in clinical trials of obesity interventions or perception studies. Exploration of these questions in larger and more diverse cohorts would be of benefit to support generalisability. Evaluation of trials that apply some of these principal findings or study, such as a greater focus on well-being and preferences around language use would be of value to confirm their efficacy in enhancing clinical trial success for the target populations.

## 5. Conclusions

Consideration of the wider impact of study design on women in obesity research needs to be considered. Our study challenges previous perceptions and suggests that addressing weight is recommended when it is done in a non-stigmatising way, while acknowledging other factors related to health such as mental and physical functioning. Clinical trialists should consider measuring well-being and using language relating to the holistic nature of health to engage women of reproductive age with obesity. Future preconception trials should consider a combination of digital and traditional recruitment strategies that resonate with altruistic motivations and use sensitive language to maximize uptake. 

## Figures and Tables

**Figure 1 ijerph-19-13832-f001:**
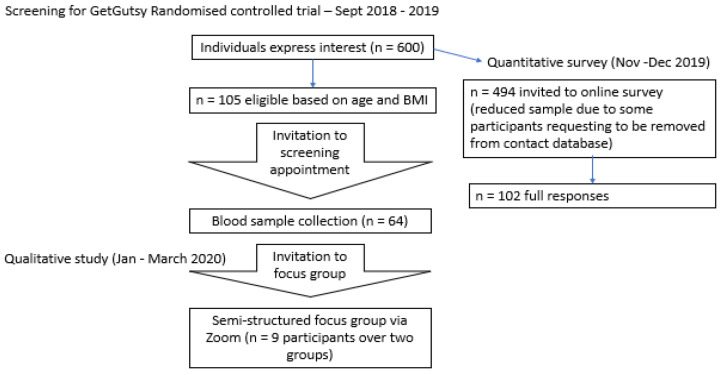
Flow diagram of the mixed-methods SWAT for the GetGutsy trial.

**Figure 2 ijerph-19-13832-f002:**
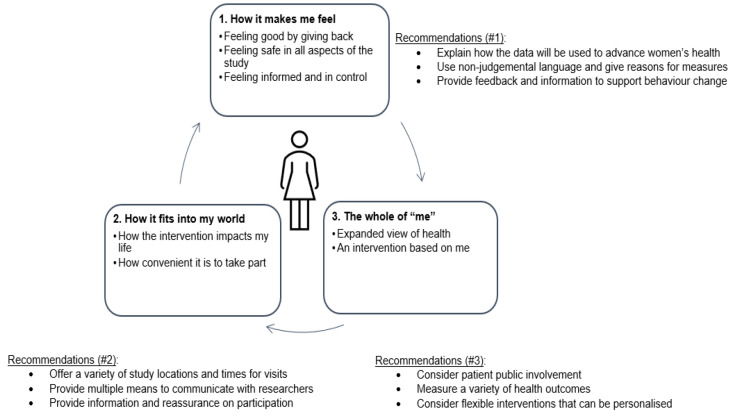
Themes and subthemes relating to taking part in research and recommendations for clinical trialists.

**Table 1 ijerph-19-13832-t001:** Characteristics of survey respondents (n = 102).

Survey Question	n (%) ^a^
Age (years)	42.0 (5.6)
BMI (kg/m^2^)	24.2 (22.3, 28.1)
Employment Full time work Part time work Student Unemployed	74 (72.5) 18 (17.6) 6 (5.9) 4 (3.9)
Education (completed third level)	79 (77.5)
No prior experience taking part in research	49 (48.5)
Recruitment into the study Heard by email Heard by ezine/newsletter Phone / messaging application Recruitment stand Told by friend or relative Told by colleague Poster at work Poster outside work Radio Browsing social media	35 (34.7) 9 (8.8) 5 (4.9) 4 (3.9) 4 (3.9) 11 (10.9) 16 (16.3) 8 (7.8) 2 (2.0) 15 (14.7)
Response time to recruitment strategy Same day A day or two later One week later Two weeks later Greater than two weeks later	38 (37.1) 43 (42.2) 12 (11.8) 2 (2.0) 7 (6.9)
Motivation for taking part I was interested in the health screen I was interested in getting the probiotic I wanted to support health research I wanted to be healthier All the above	30 (29.4) 8 (7.8) 38 (37.3) 21 (20.6) 5 (2.9)

^a^ Values are presented as n (%) except for age which is presented as mean (standard deviation) and BMI which is presented as median (interquartile range).

**Table 2 ijerph-19-13832-t002:** Characteristics of the focus group participants.

Focus Group	ID	Age (Years)	BMI Category (kg/m^2^)	Ethnicity	Education	Parity	Method of Recruitment into the Study *
FG1	P1	53	30–35	White	Some third level	2	Email at work
FG1	P2	40	30–35	White	Completed third level	0	Recruitment stand at work
FG1	P3	30	28–29.9	White	Completed third level	0	Email at work
FG1	P4	40	35–40	White	Completed third level	1	Word of mouth–family member (healthcare professional)
FG1	P5	36	35–40	White	Completed third level	0	Word of mouth–friend (healthcare professional)
FG2	P6	36	30–35	White	Completed third level	0	Spoke with member of research team at another study
FG2	P7	42	30–35	White	Completed third level	2	Responded to poster put up in local hospital
FG2	P8	42	28–29.9	White	Completed third level	0	Email at work
FG2	P9	44	35–40	White	Completed third level	3	Post on Facebook

Demographics were collected as part of the original GetGutsy trial; * Identified through content analysis of transcripts. BMI = Body Mass Index. A BMI ≥ 28 kg/m^2^ was the inclusion criteria to qualify for a blood test for an atherogenic lipid profile in the GetGutsy host trial. The BMI categories reported here are the categories in which matches measured BMI at the blood test for each participant fell.

## Data Availability

The data presented in this study are available on request from the corresponding author. The data are not publicly available due to privacy reasons.

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
