# Peer review of "Recruiting and Engaging Women of Reproductive Age with Obesity: Insights from A Mixed-Methods Study within A Trial"

_ijerph, 2022, doi:10.3390/ijerph192113832_

Round 1
Author Response
Reviewer 1
Title: Recruiting and engaging women of reproductive age with obesity: insights from a mixed-methods study within a trial – This is a well written paper and provides insights that may help future researchers.
Thank you for these comments.
– In terms of overall content these would be considerations for the authors:
o It would be helpful to acknowledge the limitations of BMI-classified obesity and the other options for diagnosis / classification in the manuscript – consider doing so in the discussion where you discuss weight and health outcomes.
This was added to the section on the host trial as follows “The original GetGutsy study included a weight criterion to identify participants with potential atherogenic lipid profile. In the recent Canadian Adult Obesity Clinical Practice Guidelines, obesity is defined as the presence of excess adiposity that nega-tively impacts health 37. Using this definition, a raised BMI alone is not considered obe-sity. Applying this definition to metabolic health alone would remove the metabolically healthy obesity phenotype, leaving only “healthy” or “obesity”. Some studies, however, suggest that there is risk of adverse outcomes in obesity based on BMI even without any evidence of metabolic derangement 38-40. Others have suggested that metabolically healthy obesity is a transient state, during which individuals with obesity based on BMI are at increased risk of developing metabolic derangements overtime 41-43. Taken together, this data suggests that regardless of metabolic status, it is im-portant that people with raised BMI have access to treatment should they wish to re-ceive it and clinical trialists should consider this in future study design 44’’
O One interesting aspect that would be worth exploring in the discussion is the evidence that people living with obesity engage less in health screening – yet that was the opposite of what you found.
Thank you for highlighting this discussion point. We agree this is an interesting finding. To address this we added the following paragraph to the discussion “We found just over 50% of survey respondents reported wanting a health screen or to be healthier as their main motivation for registering interest in the GetGutsy study. It is long documented in the literature that people living with obesity may have less engagement with health screening for a variety of factors, including weight stigma 45-48. While our survey was completed by women regardless of BMI, we also found an interest in health screening in our focus groups, which included only women with overweight or obesity. In a recent study, other factors such as education, age and loca-tion of residence were associated with missed health screenings 49. It is possible that so-ciodemographic factors explain the favourable view of health screening in our study. Our study was not about weight loss and as such, recruitment content focused on health screening including lipid analysis rather than weight. The qualitative compo-nent of this SWAT identified an interest in learning new things about health amongst women with overweight and obesity. Future clinical trials and health services can con-sider the value of novelty to frame health screenings.
O The abstract doesn’t do justice to the quality of the paper! I was confused about what an expanded view of health meant so I think that needs to be explained, and also what was meant by other health markers not being considered sensitive. It’s challenging with the word count, but I really liked the discussion opening – it’s a great summary and highlights the findings in context very well. I think the abstract would be more engaging if the opening points in the discussion were emphasised. – Then a few more specific considerations:
Thank you for this comment. We appreciate the opportunity to enhance the clarity of the abstract and have changed it to read as follows: “Engaging women with obesity in health-related studies during preconception is challenging. Limited data exists relating to their participation. The aim of this study is to explore the experiences and opinions of women participating in a weight-related, preconception trial. This is an explanatory sequential (quan-QUAL) mixed-methods Study Within A Trial, embedded in the GetGutsy randomized controlled trial (ISRCTN11295995). Screened participants completed an online survey of eight questions (single or multiple choice and Likert scale) on recruitment, motivations, and opinions on study activities. Participants with abdominal obesity (waist circumference ≥80cm) were invited to a subsequent semi-structured, online focus group (n=2, 9 participants) that was transcribed and analyzed using inductive thematic analysis, with a pragmatic epistemological approach. The survey (n=102) showed the main research participation motivations were supporting health research (n=38, 37.3%) and wanting health screening (n=30, 29.4%). Most participants were recruited via email (n=35, 34.7%) or social media (n=15, 14.7%). In the FGs, participants valued flexibility, convenience and research methods that aligned with their lifestyles. Participants had an expanded view of health that considered emotional well-being and balance alongside more traditional medical assessments. Clinical trialists should consider well-being, addressing the interconnectedness of health and incorporate a variety of research activities to engage women of reproductive age with obesity. “, word count = 204.
o Page 1, Line 42: refer to BMI classified obesity – this is rectified thank you.
o Page 1, Line 44: Maybe not just about efficacy – also consider mentioning that it’s about the quality of the intervention delivery, having representative samples, and understanding of why people engage…
Thank you for this recommendation. We agree there are other important aspects than efficacy. Please see the updated section in the following, which addresses these points. “Reaching the right population group is essential for the efficacy of health interventions and insights from these populations helps inform appropriate intervention delivery 1, 2. Systematic reviews of trials in young adults with overweight or obesity show that of-ten only small sample sizes are achieved, and this limits statistical power 7, 8. The prob-lem of low recruitment is further exacerbated by low retention rates among women of reproductive age 9. Much of the research to date on the motivations, barriers and enablers, and over-all preferences of women with overweight or obesity on study conduct has focused on weight loss or maternal health 10-12. This is an issue as weight loss or pregnancy may not be a priority for many young women with obesity, affecting their engagement with these trials 13-16. It is acknowledged that focusing on improving the health of all women of reproductive age could help to avoid these issues, yet there is a lack of data on the preferences of women in this group in relation to study participation 17. This limits our ability to design successful health intervention studies with sufficient quali-ty and representative samples 7-9.”
o Page 2, Line 52: Consider changing ‘healthier changes’ to ‘health behaviour changes’
Thank you for this suggestion. The sentence has been removed in relation to other edits.
o Page 2, Line 93: should that be ‘make health behaviour changes’?
Thank you for this suggestion. We amended to reflect this categorisation as follows and were not planning health behaviour changes aimed at weight loss in the next 3 months.
o Page 3, Line 127: Did you include / consider including people living with obesity in the survey validation? If not, consider mentioning that as a limitation / recommendation for future work (also relevant to input to the topic guide further on) o Page 3,
Thank you for highlighting this limitation. We did not include people living with obesity in the study in relation to content validity or study design. We agree that in future this would enhance the study and have added it to the limitations here: We did not include any patient or public involvement in this study, or the original GetGutsy. Future research would benefit from early involvement of patient-advocacy groups and other stakeholders to gain further insight into factors influencing study participation along with survey validation. This would enhance our confidence in the questions being asked and their acceptability and importance to the target audience.”
Line 129: Excel needs a reference
Thank you for highlighting. The version details have been added to the text as follows: Microsoft® Excel® for Microsoft 365 MSO (Version 2209 Build 16.0.15629.20152) 64-bit (Line 157).
Page 5: I’m a little confused by the BMI in the table at 24 – was the criteria not greater than this to even take part? Was this self reported? If it was do say this as it may confuse the readers
Thank you very much for highlighting this confusion. To clarify, please see this detail added to the table footnote: A BMI ≥28kg/m2 was the inclusion criteria to qualify for a blood test for an atherogenic lipid profile in the GetGutsy host trial. The BMI categories reported here are the categories in which the measured BMI at the blood test for each participant fell.
Reviewer 2 Report
dear authors
thank you for your submission
we read the submitted paper; we have a few suggestions to improve its impact
1. The introduction did not define the defect in knowledge in earlier studies
nor the rationale for choosing this subject
2. Methods
a) the flowchart is poor quality;it should be at least 300db in clarity please update
b)the number included; invited and excluded are not well presented ...in the flowchart, you said 102 responses and 105 were eligible while in the ext 107?
which is true? how was exclusion made?
c)the sample size required to perform this study is not calculated at 102 only while the FG was only 9 ?? this is too small to reflect your population?
d) the subgrouping of FG; how was it done?
e) the methods were extensively mentioned; revision is needed.
3. Results and statistics
no comments
4. discussion
the future implication of the study results in the current knowledge in the field
suggestion of future research in the topic
limitation of the study was not addressed well
REFERANCES
no comments
Author Response
Reviewer 2:
we read the submitted paper; we have a few suggestions to improve its impact
- The introduction did not define the defect in knowledge in earlier studies
nor the rationale for choosing this subject
Thank you very much for your suggestion. We agree the knowledge gap needed further clarification. This is addressed as follows “ Much of the research to date on the motivations, barriers and enablers, and over-all preferences of women with overweight or obesity on study conduct has focused on weight loss or maternal health 10-12. This is an issue as weight loss or pregnancy may not be a priority for many young women with obesity, affecting their engagement with these trials 13-16. It is acknowledged that focusing on improving the health of all women of reproductive age could help to avoid these issues, yet there is a lack of data on the preferences of women in this group in relation to study participation 17. This limits our ability to design successful health intervention studies with sufficient quality and representative samples 7-9.”
- Methods
- a) the flowchart is poor quality;it should be at least 300db in clarity please update
Thank you, we agree and have provided an updated image.
b)the number included; invited and excluded are not well presented ...in the flowchart, you said 102 responses and 105 were eligible while in the ext 107?
which is true? how was exclusion made?
Thank you for this request for clarification. 105 women out of the 600 interested met the age and BMI criteria for the original GetGutsy trial. Of these, we invited 60 to take part in the focus group. The 102 women refers to the women who completed a survey, this comes from a larger pool of women who were sent the survey. These women did not have to meet the age or BMI criteria as we wanted to capture opinions broadly as they related to a trial designed for young women with obesity. The reference to 107 was the number who tried to complete a survey but 5 missed at least one question, so we included only the 102 with full responses. This is addressed in the text (line 118-126) as follows “Women were eligible to take part in the SWAT survey if they underwent initial basic screening for the GetGutsy trial (height, weight, and age) and consented to future contact (Fig.1). As the aim of the survey was to collect broad insights on aspects of conduct for a study designed for young women with obesity, all women who expressed interest in the study were invited, even if they did not meet the criteria for study participation. As the aim of the focus groups was to gain rich data on the specific aspects of the study as they related to the target population, only women who were potentially eligible to original GetGutsy trial (met all criteria except an atherogenic lipid profile) were invited to take in the SWAT FG. “ We removed the reference to 107 for clarity.
c)the sample size required to perform this study Is not calculated at 102 only while the FG was only 9 ?? This is too small to reflect your population?
Thank you for highlighting this aspect. We did not calculate a desired sample size apriori in relation to this SWAT as our potential sample was limited to those who registered interest in, or were potentially eligible for, the original GetGutsy trial. We addressed this in line 127 “We did not limit the sample size for this study to gain the maximum size possible within the available participant pool from the original GetGutsy trial.“ and 454 “Finally, our study did not have an apriori sample size calculation as participants were already limited to those who were either interested in, or potentially eligible for, the original GetGutsy trial. While this does not negate the insights gained from this descriptive study, it is possible that further insights could be gained from a larger sample, particularly in relation to the focus groups, for which we had two in this study. Future research could build on our approach to address this.”
- d) the subgrouping of FG; how was it done?
Thank you for requesting this clarification. Women were allocated to one of two focus groups based on availability. This has been added to line 153.
- e) the methods were extensively mentioned; revision is needed.
Thank you for highlighting this consideration. The methods were written in line with COREQ checklist which necessitated detailed information around many aspects. To aid the reader however, we aimed to decrease the extensiveness of this detail throughout the updated manuscript.
- Results and statistics
no comments
- discussion
the future implication of the study results in the current knowledge in the field
Thank you for highlighting this area in need of expansion. In relation to weight as a criterion, we added the following “Our results provide further insight and nuance into recent research suggesting that there is a disconnect between the perceived offence with discussions on weight by in-tervention delivery agents and people living with obesity, including women of repro-ductive age 53,54. This suggests that trials and clinicians should consider mentioning addressing but not focusing on weight alone. This will provide people with obesity with the information they need whilst reducing weight stigma.’’.
We have also addressed this in the conclusion “The wider impact of research study design on women in obesity research needs to be considered. Our study challenges previous perceptions and suggests that addressing weight is recommended when it is done in a non-stigmatising way, while acknowledging other factors related to health such as mental and physical functioning. Clinical trialists need to consider measuring well-being and use language relating to the holistic nature of health to engage women of reproductive age with obesity. Future preconception trials should consider a mix of digital and traditional recruitment strategies that resonate with altruistic motivations and use sensitive language to maximize uptake.”.
suggestion of future research in the topic
Thank you for highlighting this area for further development. These have been alluded to throughout the discussion in the manuscript. To specifically address this, the following paragraph has been added to the end of the limitations “Future studies would benefit from incorporating SWAT protocols to provide more insight into what works in clinical trials of obesity interventions or perception studies. Exploration of these questions in larger and more diverse cohorts would be of benefit to support generalisability. Evaluation of trials that apply some of these principal findings or study, such as a greater focus on well-being and preferences around language use would be of value to confirm their efficacy in enhancing clinical trial success for the target populations.”
limitation of the study was not addressed well
Thank you for highlighting this issue. We agree it is important that the limitations of the study are well discussed. We added the following paragraph to the limitations section to address this: “We did not include any patient or public involvement in this study, or the original GetGutsy. Future research would benefit from early involvement of patient-advocacy groups and other stakeholders to gain further insight into factors influencing study participation along with survey validation. This would enhance our confidence in the questions being asked and their acceptability and importance to the target audience. Finally, our study did not have an a priori sample size calculation as participants were already limited to those who were either interested in, or potentially eligible for, the original GetGutsy trial.”
REFERANCES
no comments